# Topological solitonic macromolecules

Hanqing Zhao[1], Boris A. Malomed[2,3] & Ivan I. Smalyukh [1,4,5,6] ✉

Being ubiquitous, solitons have particle-like properties, exhibiting behaviour often associated with atoms. Bound solitons emulate dynamics of molecules, though solitonic analogues of polymeric materials have not been considered yet. Here we experimentally create and model soliton polymers, which we call "polyskyrmionomers", built of atom-like individual solitons characterized by the topological invariant representing the skyrmion number. With the help of nonlinear optical imaging and numerical modelling based on minimizing the free energy, we reveal how topological point defects bind the solitonic quasi-atoms into polyskyrmionomers, featuring linear, branched, and other macromolecule-resembling architectures, as well as allowing for encoding data by spatial distributions of the skyrmion number. Application of oscillating electric fields activates diverse modes of locomotion and internal vibrations of these self-assembled soliton structures, which depend on symmetry of the solitonic macromolecules. Our findings suggest new designs of soliton meta matter, with a potential for the use in fundamental research and technology.

Recent advances in understanding light-matter interactions and topology allow for designing various artificial media, called meta-materials, assembled of pre-engineered atom-like building blocks[1-5]. Metamaterials often help to overcome limitations of properties of the natural matter[1-4]; however, known designs target only very specific physical properties, typically in a narrow range of the parameter space. On the other hand, in many cases, solitons feature classical atom-like (alias particle-like) properties, with their stability and particle-like behaviour enabled by nonlinear effects[6-11]. Proposals for the creation of bound states of solitons, in the form of "molecules"[12,13], have attracted a great deal of interest and produced experimental demonstrations in various settings. Known examples include the Newton's cradle in non-uniform soliton chains in optical fibres, which appear as an intermediate stage in the generation of the supercontinuum[9], and "supersolitons", i.e., self-trapped collective excitations in chains of fluxons trapped by a periodic lattice of defects in a long Josephson junction[6], or in chains of matter-wave solitons[10]. Another example involves reversible chaining of topological solitons[14,15] with the help of elasticity-mediated interactions. Furthermore, two- and three-dimensional (2D and 3D) crystals assembled of topological solitons[16], such as skyrmions[17-20],

hopfions[21-23] and heliknotons[24] were reported too, with interactions also mediated by the medium hosting these solitons. While solitonic structures in colloidal ferromagnetic[25], macromolecular polymeric[26] and small-molecule nonpolar nematic media could be bound together by means of elastic interactions controlled by external fields[14-25] or with the help of photopolymerization[26], such soliton binding could be only achieved in a limited range of applied fields in the former case or implied a loss of mobility in the latter case. However, the possibility of forming soliton macromolecules or metamaterial analogues of polymers with inter-soliton binding resembling strong covalent-like chemical bonds has not been considered, while they can potentially become a basis for new forms of meta matter and promise exhibiting complex dynamic behaviours. Here we show that chirality of liquid-crystal (LC) host media helps to stabilize not only individual atom-like 2D skyrmions (or small molecule-like assemblies dubbed "twistions"[27,28]) but also their macromolecule-like complex bound states. Using 3D nonlinear optical imaging[21,24,29,30] we map spatial structures of the director field $\mathbf{n(r)}$, which describes spatial variations of the local average direction of rod-like molecular orientations, the nonpolar director $\mathbf{n} \equiv -\mathbf{n}$, revealing how individual topological solitons are inter-bonded into macromolecules with the

[1]Department of Physics, University of Colorado, Boulder, CO 80309, USA. [2]Department of Physical Electronics, School of Electrical Engineering, Faculty of Engineering, and Center for Light-Matter Interaction, Tel Aviv University, P.O.B. 39040Ramat Aviv, Tel Aviv, Israel. [3]Instituto de Alta Investigación, Universidad de Tarapacá, Casilla 7D, Arica, Chile. [4]Materials Science and Engineering Program, University of Colorado, Boulder, CO 80309, USA. [5]International Institute for Sustainability with Knotted Chiral Meta Matter (WPI-SKCM²), Hiroshima University, Higashihiroshima, Hiroshima 739-8526, Japan. [6]Renewable and Sustainable Energy Institute, National Renewable Energy Laboratory and University of Colorado, Boulder, CO 80309, USA. ✉e-mail: ivan.smalyukh@colorado.edu

help of topological point defects[31,32]. Results of numerical modelling based on the Frank-Oseen free energy are consistent with these experimental findings[21,24,30], providing insights into the spontaneous self-assembly of polyskyrmionomers. Laser tweezers allow us to robustly control linear, branched and other macromolecule-resembling architectures, allowing, in particular, to encode and store data in the spatial distribution of the skyrmion number, which is a topological invariant characterizing the topology of $\mathbf{n}(\mathbf{r})$ in terms of the second homotopy group[17,18,33]. Various types of loco-motion and internal vibrations of polyskyrmionomer structures arise, caused by asymmetric responses of $\mathbf{n}(\mathbf{r})$ to pulses of the oscillating electric field. The emergent physical behaviour of poly-skyrmionomers suggests new designs of solitonic meta matter[1–4], including soliton counterparts of active matter[34]. Beyond LCs, due to the similarity of Hamiltonians, we foresee that similar structures may emerge in solid-state chiral magnets[17,18], where they can be used in the race-track-memory data storage[34,35] and other spintronics applications[36].

## Results

### Polymer-like assembly of skyrmions by defect structures

2D skyrmions are topological solitons of order parameter fields, such as magnetization field of magnets or $\mathbf{n}(\mathbf{r})$ of LCs, low-dimensional condensed matter analogues of Skyrme solitons in high energy and nuclear physics (thus often called "baby skyrmions")[37]. Within the simplest 2D skyrmion configuration (Fig. 1a), the director $\mathbf{n}$ rotates by 180 degrees from the centre to the peripheral far-field background, while overall different 2D skyrmion structures are labelled as elements of the second homotopy group $\pi_2(\mathbb{S}^2/\mathbb{Z}_2)$. The topological structures do not allow one to smoothly transform the 2D skyrmion into a uniform state. In 3D, a 2D skyrmion tube can terminate on two $\pi_2(\mathbb{S}^2/\mathbb{Z}_2)$ point defects with opposite hedgehog charges ($\pm 1$)[38], when embedded in a uniform background setting with vertical (homeotropic) boundary conditions on confining surfaces (Fig. 1b). The so-constructed 3D structures are called torons[39]. Both 2D skyrmions and torons can be stable or meta-stable in chiral LCs, playing the role of macroscopic "atoms". For perpendicular boundary conditions in the unwound

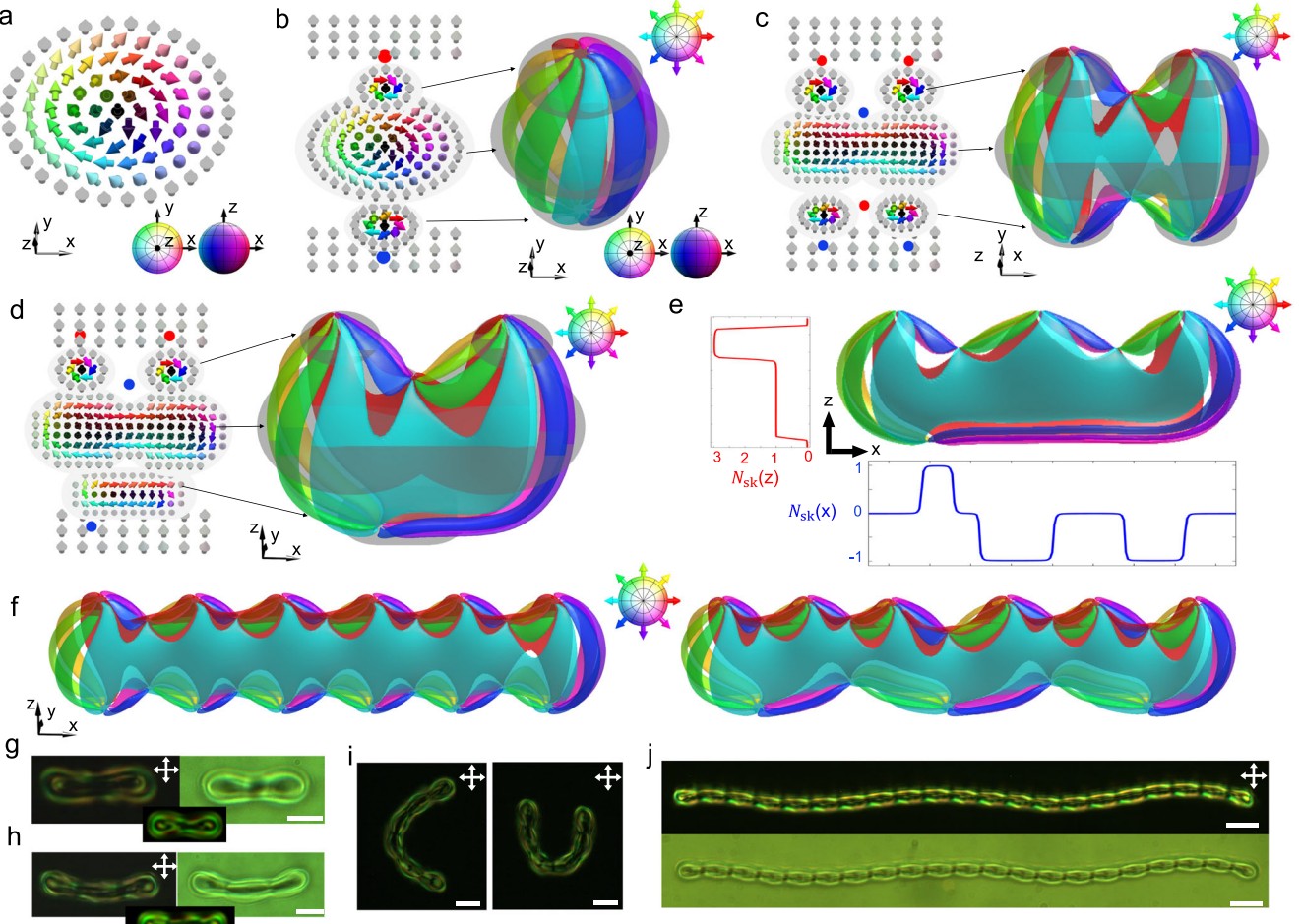

**Fig. 1 | Skyrmion, toron and polyskyrmionomer. a** Schematic of a 2D skyrmion with $\mathbf{n}(\mathbf{r})$ shown by arrows according to the coloured sphere (bottom inset) representing the order parameter space of vectorized $\mathbf{n}(\mathbf{r})$. **b** $\mathbf{n}(\mathbf{r})$ (left) and assembly of preimages (right) of a toron, shown by means of the same colouring scheme, where a preimage is the region in $\mathbb{R}^3$ (3D space) that maps to a certain single orientation on target order parameter space, and we marked the eight orientations on the top-right inset. The elementary toron comprises an internal skyrmion, capped by +1 (red sphere) and -1 (blue sphere) point defects[33]. **c** $\mathbf{n}(\mathbf{r})$ (left) and preimage assembly (right) of an $S_2^2$, which comprises two skyrmion tubes at the top, two skyrmion tubes at the bottom, and six-point defects. **d** $\mathbf{n}(\mathbf{r})$ (left) and preimage (right) of an $S_1^2$, with the defects positioned asymmetrically along the $z$

direction. **e** The distribution of the skyrmion number along the $x$ and $z$ directions of an $S_1^3$ mode, with its preimage. **f** Preimages of $S_6^6$ (left) and $S_3^6$ (right) modes. The colouring schemes in (**c–f**) for arrows and preimages are same to (**b**), and we select eight orientations as the top-right insets shown on the preimages. **g, h**, Polarizing optical micrographs (POMs) (left) and brightfield transmission-mode optical images (right) of $S_2^2$ (**g**) and $S_1^3$ (**h**) modes. Computer-simulated counterparts of the POMs are shown in the bottom insets. **i** POMs of curving hexamers, which represent word "CU" that can be robustly kept by the setting. **j** The POM and brightfield transmission-mode optical image of a polyskyrmionomer of order 21. Scale bars indicate 5 μm in (**g–i**), and 10 μm in (**j**). Crossed polarizers in POMs are shown by double arrows.

uniform background, their interaction is repulsive, therefore they cannot be bound into "molecules". However, additional opposite-charged point defects in this chiral LC system can bind the soliton "atoms" into a stable structure, effectively playing the role of shared electrons in analogy to covalent bonds (Fig. 1c). Furthermore, the diversity of possible binding scenarios includes the asymmetric structures within which the defects close to the bottom and top surfaces differ, breaking the mirror symmetry with respect to the cell midplane (Fig. 1d, e). Due to this asymmetry, the 2D skyrmion number $N_{sk}(i) = \int \boldsymbol{\Omega}_i d^2\boldsymbol{r}$ computed for 2D planar cross-sections (Fig. 1e) varies at different sample depths along the $z$ direction, where $\boldsymbol{\Omega}_i = \frac{1}{8\pi}\varepsilon^{ijk}\boldsymbol{n} \cdot (\partial_j\boldsymbol{n} \times \partial_k\boldsymbol{n})$ is the skyrmion number density along $i$ direction, $i,j$ and $k$ are spatial coordinates and $\varepsilon^{ijk}$ is the totally antisymmetric tensor[24,40]. Therefore, $N_{sk}(z)$ may be different near top and bottom confining surfaces, though it is equal to unity at the midplanes. To classify the structural diversity of such solitonic assemblies, a structure exhibiting $N_{sk}(z)=N_t$ skyrmions at the top and $N_{sk}(z) = N_b$ skyrmions at the bottom is denoted as, $S_{N_b}^{N_t}$ where "S" stands for the "skyrmion". Individual skyrmions can be assembled into a higher-order polymer-like structure (Fig. 1f), which we name "polyskyrmionomer", as said above. Much like mapping director structures from 2D planes to the two-sphere order parameter space of elementary and other skyrmions covers the two-sphere $N_{sk}$ times, $S_{N_b}^{N_t}$ describes the more complex distribution of this topological invariant within 2D planes intersecting our poly-skyrmionomers. We define the polyskyrmionomer's order $N$ as the larger number of $N_t$ and $N_b$, where polyskyrmionomer of a given order can have multiple isomers. For example, there are three types of dimers (Fig. 1g and Fig. 2): $S_2^1$, $S_2^2$ and $S_1^2$. In the experiment, these polyskyrmionomers (Fig. 1g–j and Fig. 2) can be generated by laser tweezers (Methods) under an applied voltage within a certain range.

## Data encoding in polyskyrmionomers

The 2D skyrmion number of a polyskyrmionomer not only varies along the sample depth, the $z$ direction, but also along the poly-skyrmionomer's chain extension direction, the $x$ direction (Fig. 1e and Fig. 3). For the $S_1^3$ trimer (Fig. 3a), $N_{sk}(z)=3$ in the 2D top horizontal planes parallel confining surfaces (Fig. 1e and Fig. 3b), while in the planes beneath the two point defects (Fig. 3a) the skyrmion number is reduced to unity (Fig. 1e and Fig. 3c). When computing the skyrmion number in the vertical 2D planes (parallel $z$) at different locations along the $x$ direction (Fig. 1e and Fig. 3d–f), we find that the skyrmion number density is concentrated in specific structural regions. The integration of the skyrmion number density in these localized regions yields -1/2 or +1/2, which implies the presence of merons (half-skyrmions, also referred to as twist-escaped disclinations or lambda lines)[33,41,42]. In 3D, they form meron tubes which can terminate at point defects and fade at lateral sides (Fig. 3g–i). Different combinations of positive- and negative-charge half-integer merons yield three possible values of the total skyrmion number in each $(y,z)$ cross-section (along the $x$ direction): -1, 0 and +1 (Fig. 3d–f). By changing the number and positions of point defects, multiple meta-stable states of polyskyrmionomers can be created (Fig. 3g–q), which feature different skyrmion number series along the $x$ axis. For example, $S_1^3$ configurations produce three states for different positions of the bottom negative point defect (Fig. 3g–i), whose skyrmion number series are (+1,-1,-1), (+1,-1) and (+1,+1,-1), respectively (we ignore regions with $N_{sk}(x) = 0$). These skyrmion number alternations suggest the possibility to design a scheme encoding data into the high-order polyskyrmionomers (Fig. 3r): if we treat $N_{sk}(x) = -1$ as $(0)_2$ and $N_{sk}(x) = +1$ as $(1)_2$ in the binary data presentation form, with $N_{sk}(x) = 0$ as a space separating different bits, we can use a polyskyrmionomer of order 9 to store an 8-bit binary string by arranging the specific sequence of defect positions (Fig. 3r and Supplementary Fig. 1). The standard English alphabet can be encoded by 26 different order-9 polyskyrmionomers (Supplementary Fig. 1). As an example, we show a polyskyrmionomer-based encoding of "CU" in

Fig. 3r. The polyskyrmionomer-stored information remains robust over long time because the energetic barriers associated with recon-figuring different meta-stable polyskyrmionomer structures are hundreds-to-thousands times higher than thermal energy.

## Detailed structures of polyskyrmionomers

To gain insights into the 3D $\boldsymbol{n}(\boldsymbol{r})$ structures of polyskyrmionomers, we utilize the three-photon excitation fluorescence polarizing microscopy (Supplementary Fig. 2, Methods and Supplementary Information), which we then compare with results of simulations via minimizing the Frank-Oseen free energy. Based on 3PEF-PM imaging with different polarizations of excitation laser light[29,30], the reconstructed $\boldsymbol{n}(\boldsymbol{r})$ (see Fig. 4a, b and Supplementary Movie 1) reveals point defects and localized configurations of $\boldsymbol{n}(\boldsymbol{r})$ around them (Methods). To obtain the polyskyrmionomers in computer simulations, we place the $S_1^1$ toron configurations, obtained as described previously[39], in a linear series with overlapped edges. The subsequent free energy minimization gives rise to self-compensating point defects. Within the numerical relaxation routine based on a variational method, the structure of the polyskyrmionomer relaxes to a stable state with a continuous $\boldsymbol{n}(\boldsymbol{r})$ except at the singular point defects. Asymmetric polyskyrmionomers, such as $S_1^2$ and $S_1^3$, can be obtained from the symmetric configurations, $S_2^2$ and $S_3^3$, mediated by directed annihilation of pairs of positive and negative defects (see Fig. 4a, b). The agreement of computer-simulated and experimental 3PEF-PM images in different cross-sections (see Fig. 4c, d) validates the numerical analysis of studied structures and the distribution of skyrmion numbers in our config-urations. The net charge of all $\pi_2(\mathbb{S}^2/\mathbb{Z}_2)$ point defects within the polyskyrmionomers is zero (see Fig. 4e–j), which is consistent with the uniform far-field background and boundary conditions. Spatial chan-ges of the skyrmion numbers $\pi_2(\mathbb{S}^2/\mathbb{Z}_2)$ at different 2D cross-sections orthogonal to $z$ and $x$ axes are mediated by point defects $\pi_2(\mathbb{S}^2/\mathbb{Z}_2)$. For example, while there is only one linearly stretched skyrmion in the midplane of Fig. 4f, i and Supplementary Fig. 3, this stretched skyrmion splits when the plane crosses the point defects (Supplementary Fig. 3). Imaging and modelling reveal $\boldsymbol{n}(\boldsymbol{r})$ structures comprising nonsingular configurations of the so-called lambda-lines line (see Fig. 4g–k)[33], fractional skyrmions that are pervasive in chiral nematic LCs due to their energy-costs-minimizing nature.

## Self-propelling rotation and intrinsic dynamics of the topological "molecules"

The structure and symmetry of polyskyrmionomers determine prop-erties of their driven dynamics, which can be illustrated with examples of dimers $S_1^2$, $S_2^2$ and $S_2^1$ (Fig. 1 and Fig. 2). When subjected to an oscillating electric field, these structures exhibit squirming dynamical behaviour (Fig. 5a-c) arising from their nonreciprocal asymmetric temporal evolution of $\boldsymbol{n}(\boldsymbol{r})$[14]: under the action of oscillating voltage $U$ (amplitude $U_0$ - 3.9 V), $S_1^2$ and $S_2^1$ configurations rotate clockwise and counter-clockwise, respectively. This observation is consistent with the symmetry of their 3D structures, which have broken mirror sym-metry planes passing through their middles orthogonally to both $x$ and $z$ axes. As one can obtain $S_1^2$ and $S_2^1$ from each other by upside-down flipping, as anticipated, their rotation directions are opposite. The $S_2^2$ structures remain static, except for Brownian motion, consistent with their mirror symmetry: flipping $S_2^2$ transforms it into itself. Directional rotation of asymmetric polyskyrmionomers depends on frequency of the oscillating electric field (see Fig. 5d, e): the rotation velocities of $S_2^1$ and $S_1^2$ decay exponentially and exhibit the same frequency depen-dence in opposite directions; $S_1^3$ and higher-order structures rotate slower than the lower-order ones, though yielding a similar exponen-tial decay coefficient. These high-order polyskyrmionomers also exhibit more complex dynamics, like bending, self-folding and waving, as also established for their semi-flexible chemical macromolecular analogues (see Supplementary Movie 2).

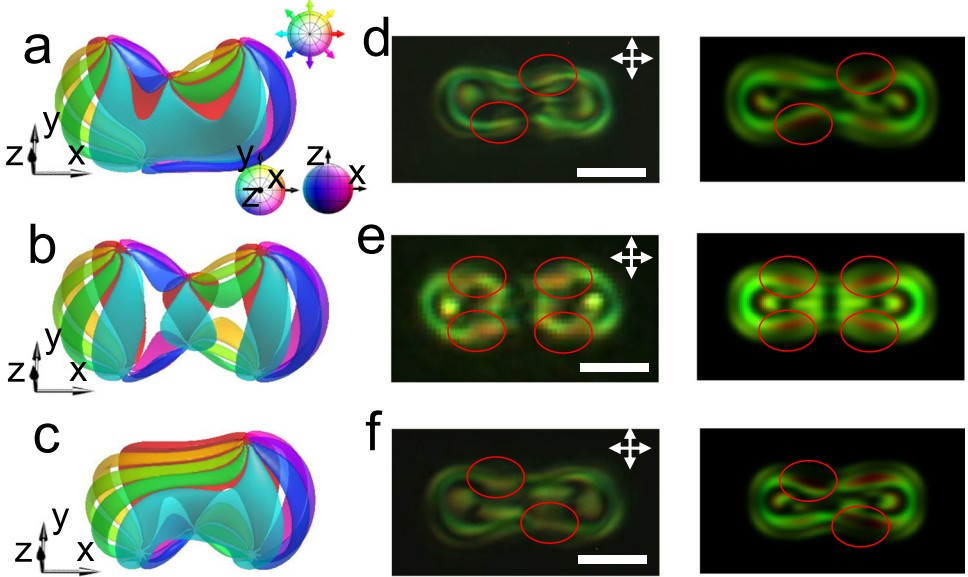

**Fig. 2 | Dimers. a–f,** Preimages (**a–c**), experimental (left) and numerically simulated (right) POMs (**d–f**) of $S_1^2$ (**a, d**), $S_2^2$ (**b, e**), and $S_2^1$ (**c, f**). Differences in the POM features are marked by red circles. The scale bars correspond to 5 μm.

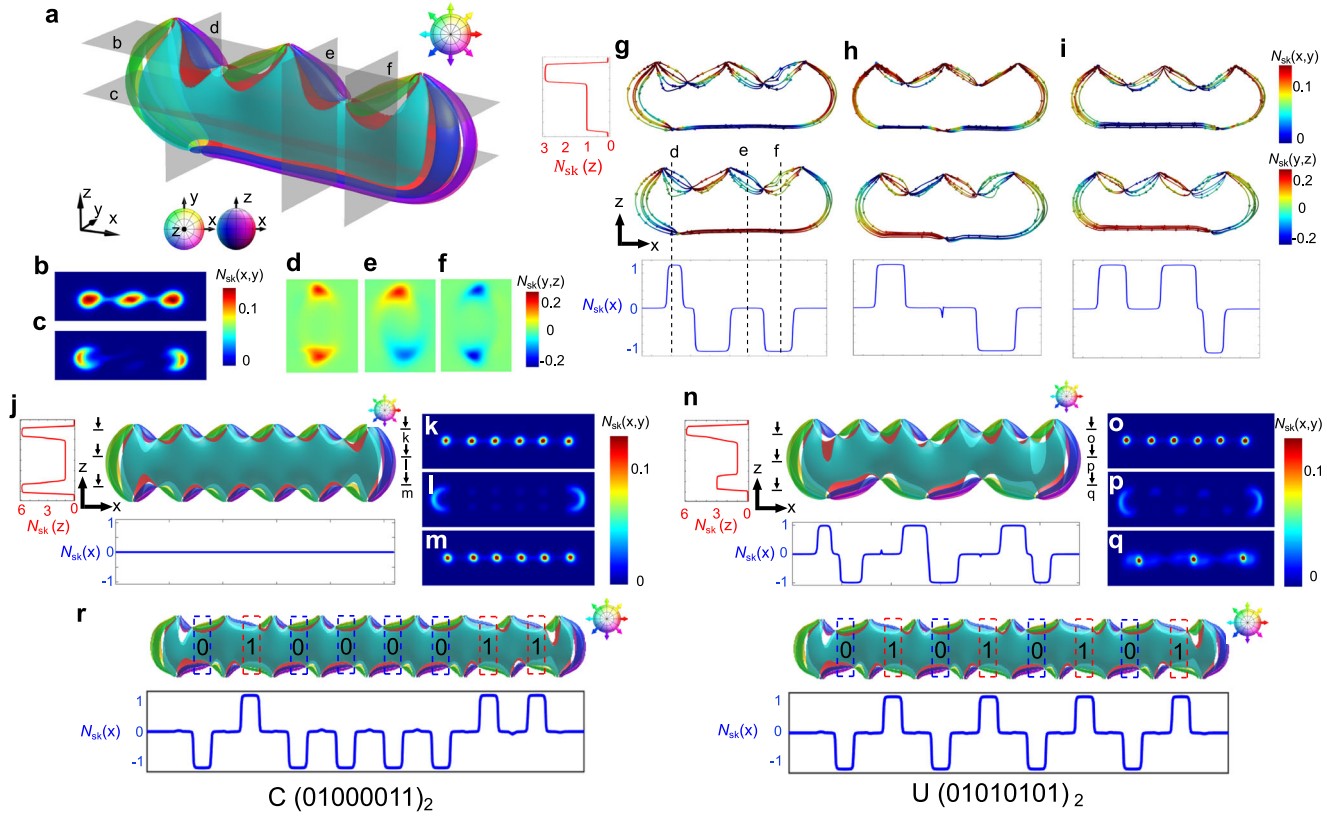

**Fig. 3 | Spatial distribution of 2D skyrmion number of polyskyrmionomers.**
**a–f** The preimage and skyrmion number density in different cross-sections of a $S_1^3$. Locations of the cross-sections corresponding to panels (**b–f**) are marked in (**a**) and the colour code used in ($x,y$) and ($y,z$) cross-sections are presented on the right. **g–i,** Streamlines of skyrmion number density (top) and skyrmion numbers along the $x$ direction (bottom) and the $z$ direction (left) for three $S_1^3$ configurations. **j** The preimage and skyrmion number distribution along the $x$ and $z$ directions of an $S_6^6$. **k–m,** The skyrmion number density in different cross-sections of the $S_6^6$ along $z$.

The locations of the cross-sections corresponding to panels (**k–m**) are marked in (**j**). **n** The preimage and skyrmion number distribution along the $x$ and $z$ directions of an $S_3^6$. **o–q** The skyrmion number density in different cross-sections of the $S_3^6$ along $z$. The locations of the cross-sections corresponding to panels of (**o–q**) are marked in (**n**). The colour codes used in (**k–m, o–q**) are presented on their right. **r,** Encoding of word "CU" using the ASCII binary table. The selected $\mathbf{n}(\mathbf{r})$-orientations corresponding to preimages are marked on the right-top, respectively.

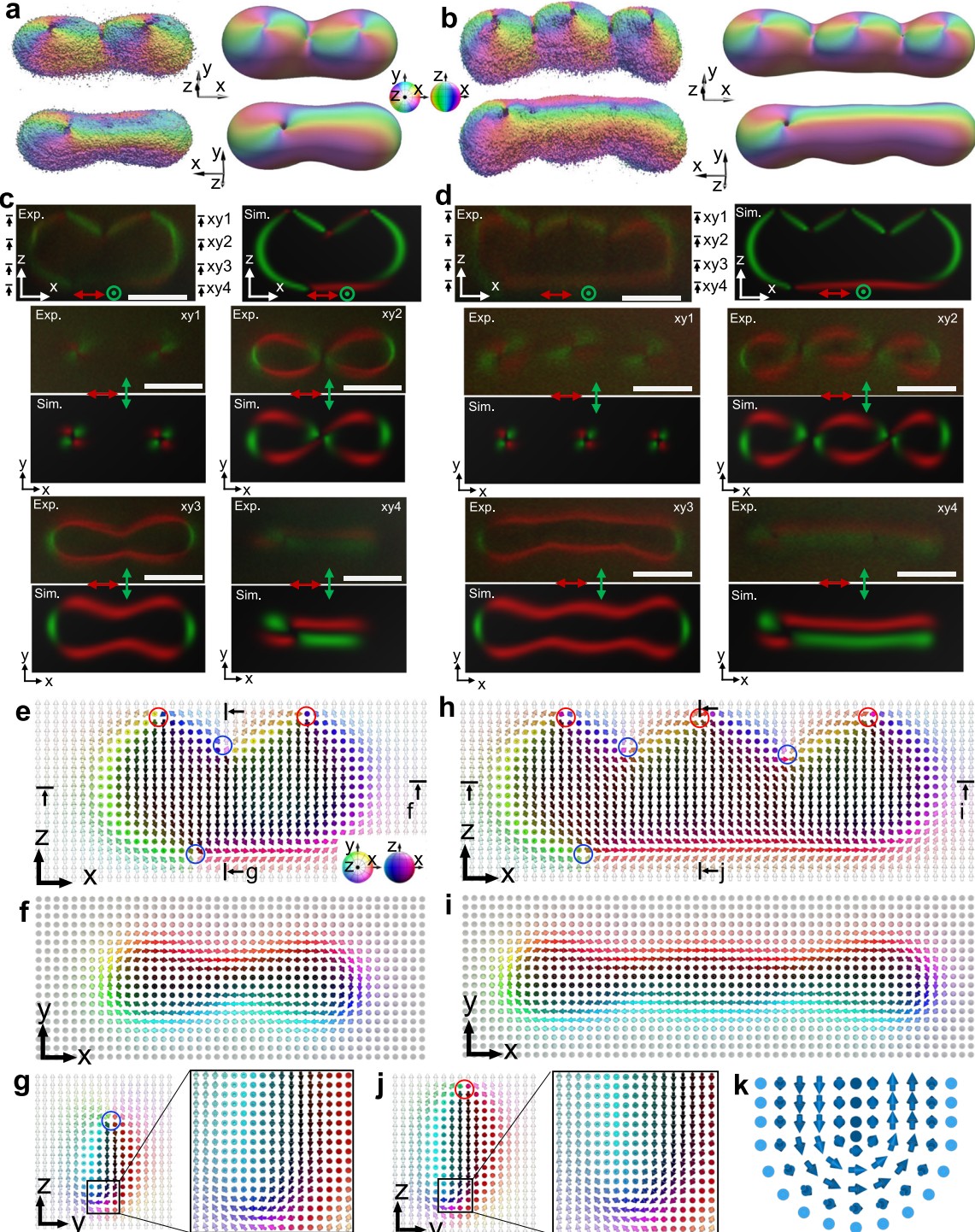

**Fig. 4 | Experimental and numerical analysis of structures of $S_1^2$ and $S_1^3$.**
**a**, **b**, Experimentally reconstructed (left) and numerically simulated (right) iso-surfaces that depicts $\mathbf{n}(\mathbf{r})$ of $S_1^2$ (**a**) and $S_1^3$ (**b**) at $|n_z| = 0.8$. The iso-surfaces of the top row are from the overhead view, and the iso-surfaces of the bottom row are from a $z$ perspective view position, as indicated by the flipped coordinate system in the inset. The iso-surface colours represent the polar and azimuthal angles of the director orientations, as defined by the coloured sphere. **c, d** Experimental (Exp.) and numerically simulated (Sim.) 3PEF-PM images of $S_1^2$ (**c**) and $S_1^3$ (**d**). The cross-sections in the ($x,z$) plane (top row) located at the midpoint of the $y$ direction, with

marked linear light polarizations (red and green). The four viewing ($x,y$) cross-sections are marked in the ($x,z$) cross-sections. Scale bars indicate the length equal to 5 μm. **e–j**, The detailed $\mathbf{n}(\mathbf{r})$ of $S_1^2$ (**e–g**) and $S_1^3$ (**h–j**) in different cross-sections plotted by coloured arrows representing $\mathbf{n}(\mathbf{r})$ orientations. Locations of the viewing cross-sections are marked in the ($x,z$) cross-sections. The red circle indicates the +1 point defect, and the blue circle indicates the -1 point defect. **k** A schematic of a nonsingular translationally invariant structure called lambda-line, which is a fractional skyrmion[31].

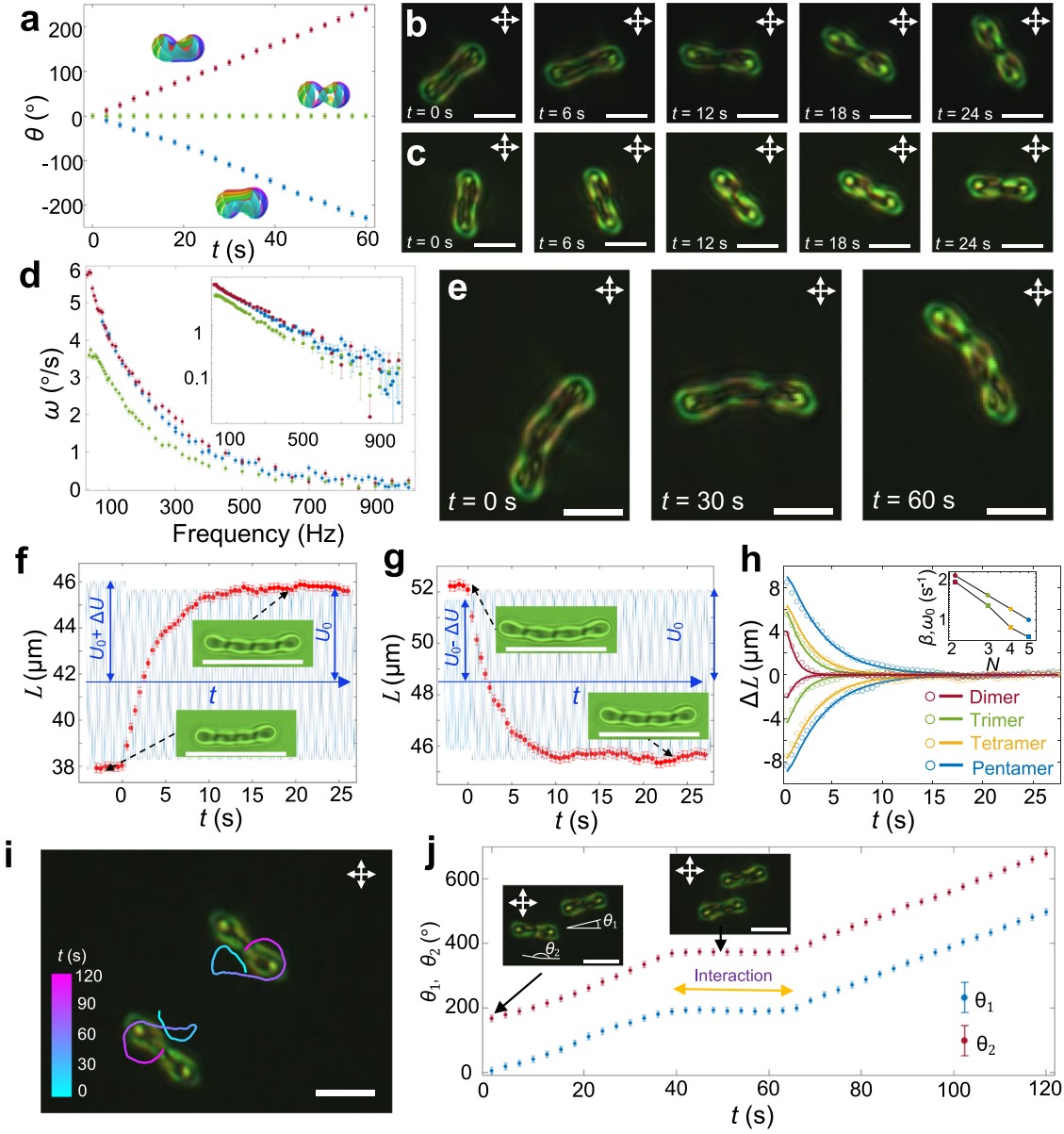

**Fig. 5 | Locomotion, vibration and dynamic interactions of poly-skyrmionomers. a** Rotation angles of $S_1^2$ (red), $S_2^2$ (green) and $S_2^1$ (blue) versus time with their preimages. The clockwise direction is defined as positive, and angles are measured with an error of $\pm 3°$. **b, c** Snapshots of self-propelling rotation of an $S_1^2$ (**b**) and $S_1^1$ (**c**). **d** Angular velocities ($\omega$) of $S_1^2$ (red), $S_1^1$ (blue) and $S_1^3$ (green) versus time. The rotation direction of $S_1^1$ is counter-clockwise, and we plot values of -$\omega$ for $S_1^1$. The semi-log-plot is shown in the inset. The errors in measuring $\omega$ are $\pm 3°$ for 200 s. **e** POM snapshots of self-propelling rotation of the $S_1^3$. **f, g** Lengths of a shrunk and extended tetramer versus time. The corresponding voltage drop and increase are plotted in blue colour ($U_0 = 3.85$ V, $\Delta U = 0.05$ V). Insets show changes of the length of the tetramers. Scale bars indicate 45.6 μm, which is the equilibrium length under the action of $U_0$. Lengths are measured with an error of $\pm 0.2$ μm. **h** Displacements of the length of the shrunk or extended dimer (red), trimer (green), tetramer (yellow) and pentamer (blue) versus time. Circles represent data measured in the experiment, and the solid lines are the best fits from the solution of the over-damped vibration, as predicted by Eq. (1). $\beta$ and $\omega_0$ are shown in the log-log-plot inset by filled circles and squares, respectively. **i,** Interaction of two dimers: the time is colour-coded on the trajectories according to the scale on the left. **j** Rotation angles of the two interacting dimers in (**i**) versus time. $\theta_1$ and $\theta_2$ are defined in the insets, which are measured with an error of $\pm 3°$. The scale bars correspond to 10 μm in (**b, c, e, i, j**).

Since skyrmions are fundamental elements of our topological molecule entities, similar to atoms, thermal fluctuations or external fields can cause various vibrations reflecting interactions between the bound quasi-atoms. The observed over-damped vibrations of poly-skyrmionomer are consistent with low Reynold number in our system (see Fig. 5f, g). The initial length $L$ of the polyskyrmionomer depends on an equilibrium-defining voltage $U$. As the amplitude of voltage $U_0$ is suddenly changed by $\Delta U$, the polyskyrmionomers shrink or extend so as to reach a new equilibrium length (see Fig. 5f–h), according to an equation of the over-damped vibrations describing the change of the length $\Delta L$:

$$\frac{d^2\Delta L}{dt^2} + 2\beta\frac{d\Delta L}{dt} + \omega_0^2\Delta L = 0, \tag{1}$$

where $\beta$ is the effective damping coefficient and $\omega_0$ is the eigen-frequency of the macrosoliton's vibrations. The best fits of solutions of Eq. (1) to the observed data for dimers, trimers, tetramers and penta-mers (Fig. 5h) demonstrates that $\beta$ and $\omega_0$ scale inversely to the order $N$. This finding is reasonable, similar to how $\beta$ and $\omega_0$ would scale for $N$

strings connected in a series, where the respective constants also decay as $1/N$. When switching $U$, the polyskyrmionomer oscillates between two equilibrium lengths (see Supplementary Movie 3) and displays breathing-like behavior (Fig. 6). Non-contact laser tweezers

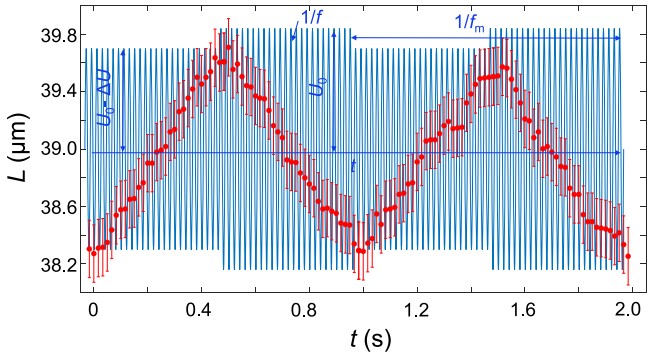

**Fig. 6 | Breathing dynamics of a trimer.** Red circles represent the length of a trimer, under the action of the modulating AC voltage (blue), versus time within two periods. The modulating frequency is $f_m = 1\,Hz$, while $f = 3000\,Hz$ ($U_0 = 3.85\,V$, $\Delta U = 0.05\,V$). The end-to-end lengths are measured with an error of $\pm 0.2\,\mu m$.

also prompt linear over-damped vibrations in the polyskyrmionomers of different modes (see Supplementary Movie 4, Supplementary Fig. 4 and Supplementary Information).

Polyskyrmionomers exhibit out-of-equilibrium inter-solitonic interactions when placed close to each other (see Fig. 5i and Supplementary Movie 5), altering directional rotations of each other (Fig. 5j) and forming an array or gas (Fig. 7a, b and Supplementary Movie 5). The gas of solitonic molecules exhibits an interesting dynamic behavior different from that of passive soliton lattices[21, 24,39,43] and other collective-motion systems[44,45]. The polyskyrmionomers can also interact with other spatially localized topological director configurations, like torons (Fig. 7c, d), hopfions (Fig. 7e, f) and möbiusons[46] (Fig. 8). When embedded in a gas-like disordered states of polyskyrmionomers (see Supplementary Movie 5), the solitonic dimers can collide and interact with torons or hopfions[33], yielding motion trajectories dependent on the size and respective viscous drag resistance to motions of the solitons experiencing these interactions. If we embed a small möbiuson[46] near a dimer, the interaction leads to their co-locomotion (Fig. 8a) while the dimer rotates with a time-varying angular velocity (Fig. 8b). This co-propagation reveals the polyskyrmionomer's potential in implementing transportation of topological cargo.

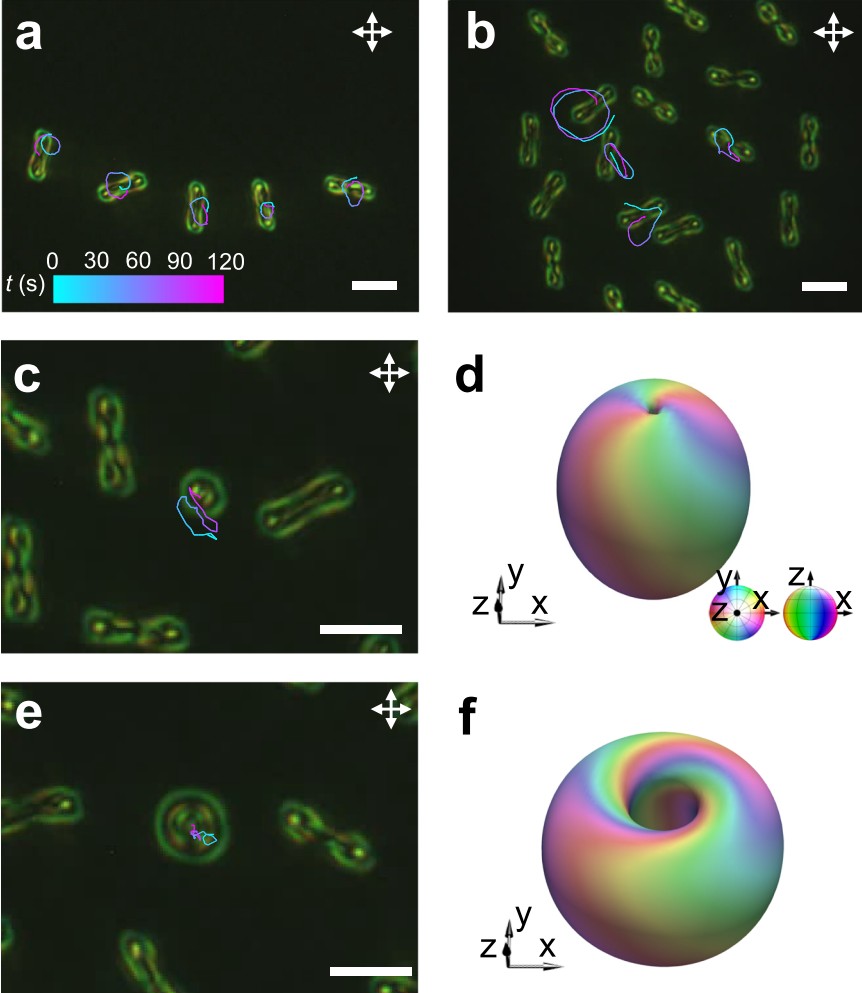

**Fig. 7 | The interaction of polyskyrmionomers with other solitons. a** A dimer array. **b** A dimer gas. **c** A toron interacting with dimer gases. **d** Numerically simulated iso-surface that depicts the 3D $\mathbf{n}(\mathbf{r})$ of a toron at $|n_z| = 0.8$. The iso-surface colours represent the polar and azimuthal angles of the vectorized director orientations, as defined by the coloured sphere. **e** A hopfion interacting with the solitonic dimers. **f** Numerically simulated iso-surface that depict $\mathbf{n}(\mathbf{r})$ of a hopfion at $|n_z| = 0.8$. Trajectories of the geometric centre in (**a–c, e**) are plotted versus time by means of the colour-coded scale displayed in (**a**). Scale bars correspond to 10 μm and $f = 100\,Hz$.

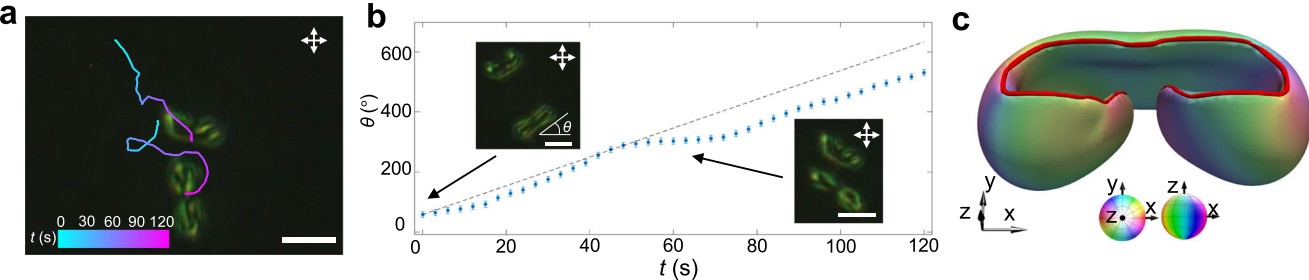

**Fig. 8 | The co-propulsion of polyskyrmionomer and möbiuson. a** POM of the interacting dimer and small möbiusson. Trajectories of the geometric centre are plotted versus time by means of the colour-coded scale displayed in the left bottom. **b** The rotation angle versus time for the interacting dimer. The insets are POM snapshots at different times and the dashed line indicates the angle versus time for a freely rotating dimer. The errors in measuring $\theta$ are $\pm 3°$. **c** Numerically simulated iso-surface that depicts $\mathbf{n(r)}$ of a möbiuson at $|n_z| = 0.8$. The vortex line is shown with red colour. The iso-surface colours represent the polar and azimuthal angles of the director orientations, as defined by the coloured sphere. Scale bars correspond to 10 µm and $f = 100$ Hz.

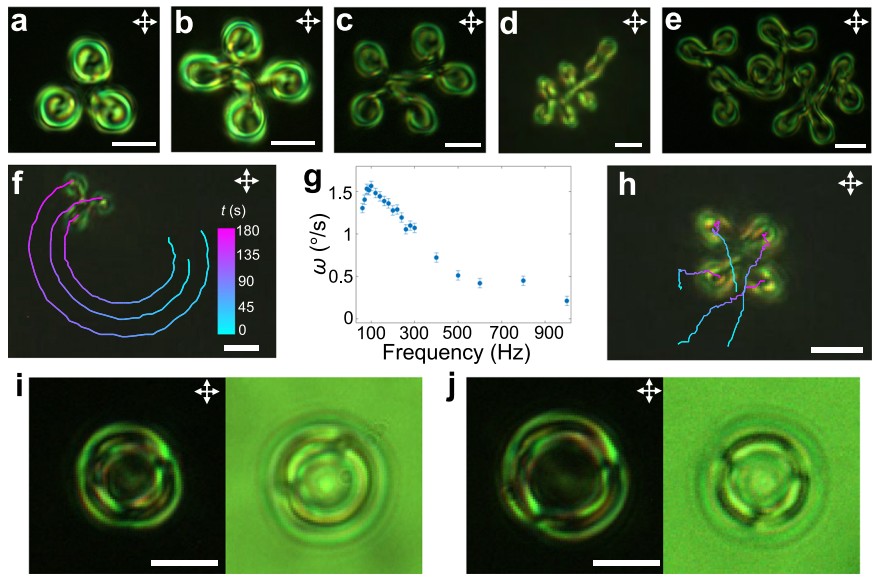

**Fig. 9 | Branched and cyclic polyskyrmionomer architectures. a–e** POMs of a star-shaped trimer, tetramer, pentamer, heptamer and oligomer. **f** Trajectories tracing three ends of a moving star-shaped trimer. **g** Rotation velocity of the geometrical centre of the trimer in (**f**) versus frequency. The errors in measuring $\omega$ are $\pm 3°$ for 200 s. **h** Trajectories tracing four ends of a moving star-shaped tetramer. The time is colour-coded on the trajectories according to the scale in (**f**) and $f = 100$ Hz. **i, j** POMs (left) and brightfield transmission-mode optical images (right) of two ring-shaped polyskyrmionomers. The amplitudes of voltage are 3.9 V in (**a–e**) and 4.0 V in (**i, j**). The scale bars correspond to 10 µm.

## Polyskyrmionomer architectures

In addition to the linearly shaped polyskyrmionomers, we have found star- and ring-shaped ones (Fig. 9), analogues of corresponding macromolecules. The star-shaped polyskyrmionomers exhibit both directional motion and rotation (see Fig. 9f–h and Supplementary Movie 6), with peaks of the angular velocity of a trimer at $f \sim 100$ Hz. Ring-shaped polyskyrmionomers can adopt different forms (Fig. 9i, j) at applied stabilizing voltage, with the voltage stability range of which being larger than for the linearly and star-shaped polyskyrmionomers. Polyskyrmionomers with different shapes can inter-transform outside of the range of stabilizing voltage. In particular, linearly shaped polyskyrmionomers transform from higher-order to lower-order ones with the increase of voltage ($\Delta U > 0.2$ V), which is accompanied by a progressive decrease of the length and discrete decrease of the skyrmion number measured in specifically selected planes (Fig. 10 and Supplementary Movie 7), in agreement with the simulation results. Furthermore, the ring-shaped polyskyrmionomer can shrink and transform into linear ones (Supplementary Movie 8), and the star-shaped polyskyrmionomers can transform into linear ones or monomers ($S_1^1$). Such processes of bringing the applied voltage outside the stability range to cause splitting of polyskyrmionomers could be considered being analogues of electrolytic splitting of conventional molecules[47].

## Discussion

We have reported creation of polyskyrmionomers, solitonic analogues of polymers, which are composed of point-defect-bound LC skyrmions. Much like in molecules comprising atoms bound by covalent bonds, while sharing electrons, polyskyrmionomers form via individual skyrmions sharing singular point defects. Such solitonic configurations are observed in highly technological chiral LCs, exhibiting facile reconfiguration and emergent dynamics activation with weak external stimuli. For example, high-order polyskyrmionomers exhibit self-interaction and folding in response to low-frequency voltage and laser tweezers (Supplementary Movie 9), with a striking resemblance of conventional polymer chain folding. This observation may help establish solitonic structures in chiral LCs as model systems to study not only the topological solitons in various experimentally inaccessible physical systems, but also to use them to model behavior of polymers and both equilibrium and active matter systems made from semiflexible chain-like building blocks. The fact that the LC's director is also

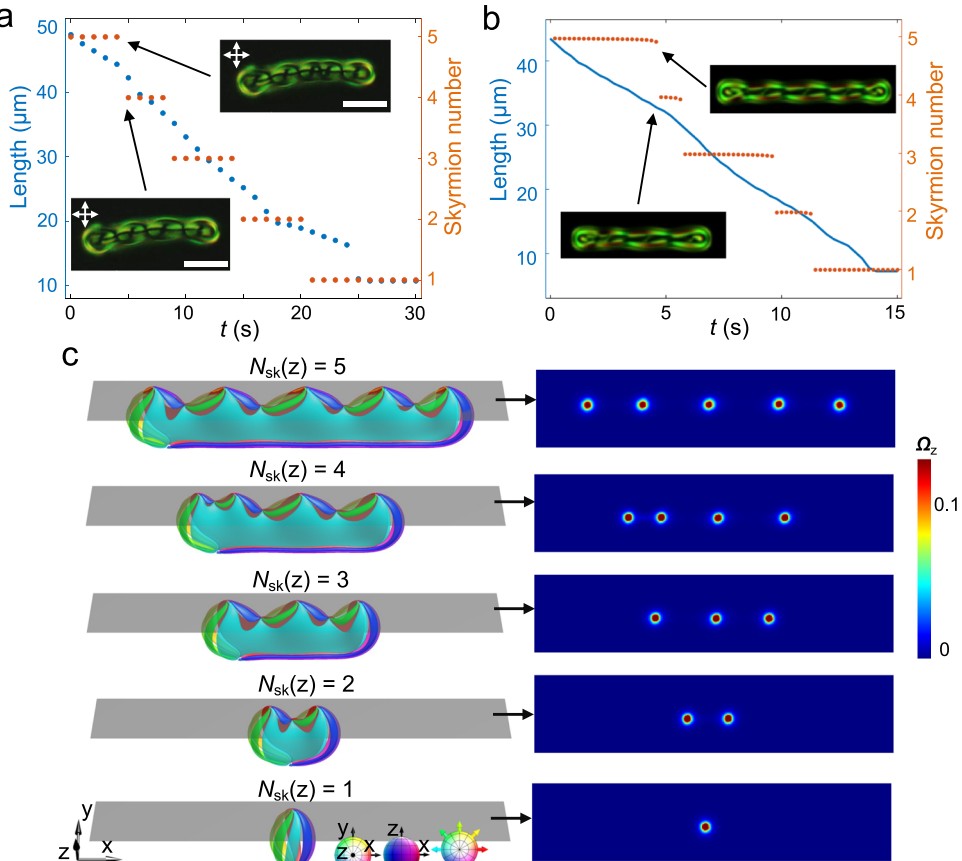

**Fig. 10 | The order annihilation process. a** Experimentally measured length and skyrmion number versus time under the action of $U_0 = 4.0$ V, starting from a pentamer. Lengths are measured with an error of $\pm 0.2$ µm. **b** The numerically simulated length and skyrmion number versus time, starting from a pentamer at $U = 2.0$ V. Insets show experimentally observed and simulated POMs before and after the annihilation. **c** Preimages and skyrmion number density in the marked cross-section along $z$ at different times. The integration of skyrmion number density corresponds to the skyrmion number marked on the left. The colour code used in the preimages and the selected orientations of preimages are marked on the bottom, the colour code used in the skyrmion number density is presented on the right. The frequency $f = 3000$ Hz and scale bars correspond to 10 µm.

the optical axis in these birefringent soft matter systems will allow designing such spatially localized structures as versatile reconfigurable beam deflectors, steerers[5] and lasing elements[48] and microcargo transporters[49]. Since the director field of all observed configurations can be smoothly vectorized, they can also emerge as stable or metastable configurations in the solid-state chiral magnetic systems with similar Hamiltonians[50]. Thus, our polyskyrmionomer architectures have potential for data storage in their magnetic nanostructured analogues[51]. In fact, junctions of skyrmion tubes with point defects, which are pervasive within the polyskyrmionomers, are also found in magnetic solids, albeit in different structures[18]. For example, similar to skyrmion bags[50], polyskyrmionomers could allow for new breeds of high-density multibit magnetic racetrack memories. From a fundamental standpoint, interesting open questions include how polyskyrmionomer structures behave in presence of flows[52] or during collective motions[44, 45].

## Methods
### Materials and sample preparation
Chiral LCs are prepared by mixing 4-Cyano-4'-pentylbiphenyl (5CB, EM Chemicals) or a nematic mixture E7 (Slichem) with left-handed chiral additive cholesterol pelargonate (Sigma-Aldrich) at a weight fraction $C_{dopant} = 1/(h_{htp}p)$ to define the equilibrium pitch ($p$) of the ensuing chiral LCs, where the helical twisting power $h_{htp} = 6.25$ µm$^{-1}$ for cholesterol pelargonate. In addition, to provide high-resolution, aberration-free 3D nonlinear optical imaging, a partially polymerizable

chiral LC mixture is prepared by mixing E7-based chiral LC (at fraction of 69%), reactive mesogens RM-82 and RM-257 (at fractions of 12% and 18%, respectively, Merck), and ultraviolet-sensitive photoinitiator Irgacure 369 (at fraction of 1%, Sigma-Aldrich). The samples are prepared by sandwiching LCs between indium-tin-oxide (ITO)-coated glass slides or coverslips treated with 25% solution of polyimide SE5661 (Nissan Chemicals) to obtain perpendicular boundary conditions. The polyimide is applied to the substrates by spin-coating at 2700 rpm for 30 s followed by a 5 min prebaking at 110 °C and 1 h baking at 185 °C. The cell gap thickness $d = 10$ µm is defined by silica spacers and the cell gap to pitch ratio is $d/p = 2$. Metal wires are attached to ITO glasses and connected to an external voltage supply (GFG-8216A, GW Instek) or a data acquisition board (NIDAQ-6363, National Instruments) for electric control. Additionally, we use an in-house MATLAB code controlling the data acquisition board, connected to a computer for fast modulation of the voltage output.

### Generating and imaging localized structures
We utilize a ytterbium-doped fibre laser (YLR-10-1064, IPG Photonics, operating at 1064 nm) to generate and non-contact manipulate polyskyrmionomers. The low-order polyskyrmionomers (linearly, star- and ring-shaped ones) are laser-generated in a uniform unwound background under the action of sinusoidal electric field (we tune the voltage to the balance value), with the laser power in the range of 40−50 mW. For higher-order linearly shaped polyskyrmionomers, we first turn on the laser, move the sample stage linearly and melt a strip

region. When the LCs quench back, a high-order linear poly-skyrmionomers emerge simultaneously. Torons, hopfions and möbiusons are also generated by laser tweezers, following the previously reported procedures[14,21,46].

POMs and brightfield transmission-mode optical images are obtained with a multi-modal imaging setup built around an IX-81 Olympus inverted microscope (which is also integrated with the 3PEF-PM imaging setup described below) and charge-coupled-device cameras (Grasshopper and Flea FMVU-13S2C-CS, Point Grey Research). High-numerical-aperture (NA) Olympus objectives 100×, 40× and 20× with NA = 1.4, 0.75 and 0.4, respectively, are used. The angular velocities and trajectories of solitons are analysed using freeware (ImageJ) from National Institutes of Health.

### Numerical modeling

For chiral nematic LCs, the energy cost of spatial deformations of $\mathbf{n(r)}$ can be expressed by the Frank-Oseen free energy functional:

$$F_{\text{elastic}} = \int \mathrm{d}^3\boldsymbol{r} \left\{ \frac{K_{11}}{2}(\nabla \cdot \boldsymbol{n})^2 + \frac{K_{22}}{2}[\boldsymbol{n} \cdot (\nabla \times \boldsymbol{n})]^2 + \frac{K_{33}}{2}[\boldsymbol{n} \times (\nabla \times \boldsymbol{n})]^2 + \frac{2\pi K_{22}}{p}\boldsymbol{n} \cdot (\nabla \times \boldsymbol{n}) \right\}.$$

(2)

Here the Frank elastic constant $K_{11}$, $K_{22}$ and $K_{33}$ determine the energy cost of splay, twist and bend deformations, respectively. Further, the surface energy is

$$F_{\text{surface}} = -\int \mathrm{d}^2\boldsymbol{r} W (\boldsymbol{n_0} \cdot \boldsymbol{n})^2,$$

(3)

where $W$ is the surface anchoring strength and $\boldsymbol{n_0}$ the preferred orientation of which is perpendicular to the surface. When external electric field $\boldsymbol{E}$ is applied, the dielectric properties of LCs induce an additional dielectric coupling term in the free energy, so that the free energy is supplemented by the following electric coupling term:

$$F_{\text{electric}} = -\frac{\varepsilon_0 \Delta\varepsilon}{2} \int \mathrm{d}^3\boldsymbol{r} (\boldsymbol{E} \cdot \boldsymbol{n})^2,$$

(4)

where $\varepsilon_0$ is the vacuum permittivity, and $\Delta\varepsilon$ is the dielectric anisotropy of the LC. The total free energy $F$ is the sum of $F_{\text{elastic}}$, $F_{\text{electric}}$ and $F_{\text{surface}}$. Polyskyrmionomers of different orders emerge as local or global minima of $F$, and a relaxation routine based on the variational method is used to identify an energy-minimizing configuration $\mathbf{n(r)}$. Applying this method, at each iteration of the numerical simulation $\mathbf{n(r)}$ is updated based on a formula derived from the Euler-Lagrange equation, $\mathbf{n}_i^{\text{new}} = \mathbf{n}_i^{\text{old}} - \frac{\text{MSTS}}{2}[F]_{\mathbf{n}_i}$, where subscript $i$ denotes spatial coordinates, $[F]_{\mathbf{n}_i}$ denotes the functional derivative of $F$ with respect to $\mathbf{n}_i$, and MSTS is the maximum stable time step of the minimization routine, determined by the elastic constants and the spacing of the computational grid. To scale the time step in the real system, we assume that the director dynamics is governed by the balance equation: $[F]_{\mathbf{n}_i} = -\gamma \frac{\partial \mathbf{n}_i}{\partial t}$, where $\gamma$ is the rotational viscosity. The end-of-the-relaxation condition is identified by monitoring the change in the spatially averaged functional derivatives in consecutive iterations. Approaches zero, it signifies proximity of the system to a steady state, and the relaxation routine comes to a halt. The 3D spatial discretization is performed on large 3D square-periodic $40N \times 40 \times 40$ grids, and the spatial derivatives are calculated using finite-difference methods with the second-order accuracy, which allows us to minimize discretization-related artifacts in modeling polyskyrmionomers of different orders. To encode data in the 8-bit binary format, we first derive stable structures of nonamers, then we slightly move the initial position of the intrinsic defects and let them relax (Supplementary Fig. 1). As a result, the non-zero skyrmion number spontaneously emerges along the $x$ direction. For all simulations, the following parameters were used: $d/p = 2$, $K_{11} = 6.4 \times 10^{-12}$N, $K_{22} = 3 \times 10^{-12}$N, $K_{33} = 10 \times 10^{-12}$N, $U = 1.9V$, $\gamma = 77$mPas and $W = 10^{-4}$Jm$^{-2}$.

The POM images are simulated by means of the Jones-matrix method, using the energy-minimizing configurations of $\mathbf{n(r)}$ for the polyskyrmionomers. We first split the cell into 40 thin sublayers along the $z$ direction, then the Jones matrix for each pixel in each sublayer is calculated by identifying the local optical axis and ordinary and extraordinary phase retardation. The optical axis is aligned with the local molecular direction, while the phase retardation originates from the optical anisotropy of the LC ($n_o = 1.58$ and $n_e = 1.77$ for 5CB). The Jones matrix for the whole LC cell is obtained by multiplying all Jones matrices corresponding to each sublayer. The simulated single-wavelength POM is obtained as the respective component of the product of the Jones matrix and the incident polarization. To properly reproduce the experiment POMs, we produced images separately for three different wavelengths spanning the entire visible spectrum (450, 550, and 650 nm) and then superimposed them, according to light source intensities at the corresponding wavelengths.

### Three-dimensional nonlinear optical imaging

3D nonlinear optical imaging of polyskyrmionomer structures is performed using the 3PEF-PM setup built around the IX-81 Olympus inverted optical microscope (see Supplementary Information). To reduce material birefringence that leads to the 3PEF-PM imaging artifacts, before imaging we applied the weak ultraviolet light to photopolymerize the polyskyrmionomers which are stabilized in the partially polymerizable chiral LC mixtures; then we washed away and replaced the unpolymerized LCs by infiltrating an index-matching immersion oil. We use a Ti-Sapphire oscillator (Chameleon Ultra II; Coherent) operating at 870 nm with 140-fs pulses at an 80 MHz repetition rate, as the source of the laser excitation light. An oil-immersion 100× objective (NA = 1.4) is used to collect the fluorescence signals, which are detected by a photomultiplier tube (H5784-20, Hamamatsu) after a 417/60 nm bandpass filter. With a third-order nonlinear process, the LC molecules are excited via the three-photon absorption process and the signal intensity scales $\propto \cos^6\beta_0$, where $\beta_0$ is the angle between the polarization of the excitation light and the long axis of the LC molecule. Different polarization states of the excitation are controlled by a half-wave plate. When $\mathbf{n(r)}$ is nearly parallel to the polarization of the laser beam, large $\cos\beta_0$ corresponds to the strong 3PEF-PM signal intensity. Then, the experimental 3D fluorescence images are processed by applying contrast enhancement and depth-dependent intensity correction. To reconstruct the polyskyrminomer structures experimentally, we define the director with respect to polar angle $\alpha$ and azimuthal angle $\varphi$, where $\mathbf{n(r)} = [\sin\alpha\cos\varphi, \sin\alpha\sin\varphi, \cos\alpha]$. We calculate the Stokes parameters using four 3PEF-PM images with the excitation laser beam linearly polarized at four different angles with respect to the $x$ axis ($I_0$, $I_{\pi/4}$, $I_{\pi/2}$, $I_{3\pi/4}$), the angle being zero for component $I_0$. Then, the appropriately defined Stokes parameters are

$$\Psi_0 = (I_0 + I_{\pi/4} + I_{\pi/2} + I_{3\pi/4})/2,$$

$$\Psi_1 = I_0 - I_{\pi/2},$$

$$\Psi_2 = I_{\pi/4} - I_{3\pi/4}.$$

(5)

Using these parameters, we calculate the fields $\alpha(\mathbf{r})$ and $\varphi(\mathbf{r})$:

$$\sin^6\alpha \propto \Psi_0/A,$$

$$\tan 2\varphi = \Psi_1/\Psi_2,$$

(6)

where *A* is the amplitude of the normalized 3PEF-PM signal. The ensuing iso-surfaces are visualized in perspective views with the help of an open-source freeware, PARAVIEW. Computer simulations of the 3PEF-PM images are also based on the signal intensity. The 3D structures, produced from both experimental and numerical data, for different linear polarizations of excitation light are coloured as shown in Figs. 4a,b, and 2D cross-sections are colourd by red and green for $I_0$ and $I_{\pi/2}$, respectively (Fig. 4c, d).

## Data availability

The data generated in this study are provided in the Source Data file. All other data that support the plots produced in this paper, as well as other findings reported in this study, are available from the corresponding author upon request. Source data are provided with this paper.

## Code availability

The codes used for the numerical calculations are available upon request.

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

## Acknowledgements

We thank T. Lee, J.-S. Tai, J.-S. Wu and P.J. Ackerman for useful discussions and technical assistance. I.I.S. acknowledges hospitality of the International Institute for Sustainability with Knotted Chiral Meta Matter (SKCM$^2$) at Hiroshima University and the Biosoft Centre at Tel Aviv University during his sabbatical visits, where he was partly working on this article. This research was supported by the U.S. Department of Energy, Office of Basic Energy Sciences, Division of Materials Sciences and Engineering, under contract DE-SC0019293 with the University of Colorado at Boulder. The work of B.A.M. was supported, in part, by the Israel Science Foundation through grant No. 1695/22.

## Author contributions

H.Z. performed experiments and numerical modeling, under supervision of I.I.S. H.Z. and B.M. performed analytical modelling. I.I.S. initiated and directed the research. H.Z. and I.I.S. wrote the manuscript, with feedback and contributions from all authors.

## Competing interests

The authors declare no competing interests.
