## [Peer Review File · Nature Communications]

REVIEWERS' COMMENTS

Reviewer #1 (Remarks to the Author):

In this study, researchers investigate the possibility of forming soliton macromolecules or metamaterial analogs of polymers with inter-soliton binding resembling strong covalent-like chemical bonds. The researchers demonstrate that the chirality of liquid-crystal (LC) host media helps stabilize individual 2D skyrmions and their macromolecule-like complex bound states. By mapping the spatial structures of the director field, which describes local molecular orientations, the researchers reveal how individual topological solitons are bonded into macromolecules using topological point defects.

Using laser tweezers, the researchers demonstrate robust control over the linear, branched, and other macromolecule-resembling architectures of polyskyrmionomers. They show that data can be encoded and stored in the spatial distribution of the skyrmion number, a topological invariant characterizing the director field's topology. The researchers propose that polyskyrmionomers could be used to design solitonic metamaterials, including counterparts of active matter.

The study also explores the possibility of similar structures emerging in solid-state chiral magnets, which could find applications in data storage and spintronics. The researchers suggest that polyskyrmionomers could be used to encode data in the high-order structures, and they present a scheme for encoding binary data using the skyrmion number distribution.

The detailed structures of polyskyrmionomers are investigated using three-photon excitation fluorescence polarizing microscopy (3PEF-PM) and numerical simulations based on minimizing the Frank-Oseen free energy. The experimental and simulated results are in agreement, validating the analysis of the structures and the distribution of skyrmion numbers.

The dynamics of polyskyrmionomers are also explored, showing self-propelling rotation and intrinsic vibrations. The asymmetry of the structures leads to nonreciprocal temporal evolution, causing the structures to rotate in different directions under an oscillating electric field. Higher-order polyskyrmionomers exhibit more complex dynamics, resembling bending and waving motions.

Overall, this study demonstrates the potential of soliton macromolecules or polyskyrmionomers as a new form of metamaterials. The unique properties and behaviors of these structures could open up

new possibilities in designing solitonic meta matter and exploring applications in various fields. For this reason I think that the paper must be accepted for publication in its current form.

Reviewer #2 (Remarks to the Author):

The paper "Topological solitonic macromolecules" reports very interesting findings on the formation and dynamics of the solitons (skyrmions) in liquid crystals. The paper comprises nicely the experimental and theoretical results that match each other perfectly. The reported results look valid and are sufficiently well explained. The focus point of the paper is the formation of complex clusters of skyrmions. Indeed, this problem is of great interest for the scientists working in the area of soliton dynamics. I agree with the authors that liquid crystals seem to be a very good test bench to study the soliton clusters. I find the idea to couple the individual skyrmions by point defects to be elegant and quite promising (though in my opinion the analogy between these defects and the shared electrons is quite remote). The "polyskyrmionomers" studied in the paper are very stable and can be manipulated. This, potentially, opens a way to use them for information coding and processing. I believe that the paper is a significant step forward in the physics of nonlinear waves. The reported effects are of interest not only for the researches working in the field of liquid crystals or magnetic systems but for wider physical community. Thus I think that the paper can be published in Nature Communications. However I have a number of critical remarks listed below.

1) The authors make the statement that "However, the possibility of forming soliton macromolecules or metamaterial analogues of polymers with inter soliton binding resembling strong covalent-like chemical bonds has not been considered, while they can be a basis for new forms of meta matter." I suggest to explain in more details why this case is of interest and if there is a fundamental difference between this kind of inter-soliton coupling and the known ones.

2) There is a statement that equation (1) describes over-damped oscillations. If the oscillations are over-damped then, probably, the equation can be reduced to the first order equation describing the relaxation of the system to the equilibrium state.

3) This question is related to the previous one. Can equation (1) be understood as the equation for the amplitude of a mode (in the context for the lowest mode amplitude) of the coupled beads described by something like

$$\ddot{x}_n + \gamma \dot{x}_n + \sigma(2x_n - x_{n+1} - x_{n-1}) = 0 ?$$

Then, indeed, the eigenfrequency of the lowest mode scales as $1/N$ where N is the number of the beads. However I do not quite understand the reason why this scaling works for γ .

4) I believe that it has to be explained in the text what is the difference between gas-like and liquid-like states (lines 192-200)

5) I have to complain on the quality of the figures. The important elements are often so small that they can barely be seen. For example in Fig.1 it is difficult to understand where some arrows are directed to. Also, I would say that the figures contain too many panels; they are very densely packed, messy and difficult to read. I believe that the figures captions can also be improved.

Reviewer #3 (Remarks to the Author):

The manuscript explores the properties of the soliton polymers that authors called a polyskyrmionomers, which, similarly to optical skyrmions, can be characterized by the skyrmion number. The authors present the ubiquitous properties of polyskyrmionomers, such as data encoding by the spatial distribution of the skyrmion number. The presented research is novel and confirmed by the series of high-quality experimental verifications, like 3D nonlinear imaging, undertaken to present the director's spatial distribution in analyzed samples. Also noteworthy is the achievement of high agreement with numerical simulations. This interesting work fosters a better understanding of light-matter interactions in complex pre-engineered optical structures. The manuscript is generally well-written. I think that this work has the potential to be published in Nature Communications Journal.

We thank the Referees for their positive reviews. Following comments and suggestions put forward by the Referees, we have revised the manuscript. We would like to thank the Referees for their valuable comments. We also supply all the files prepared for the final submission, according to your guidelines.

Changes to Account for the Referees' Suggestions.

Report of Referee 1:

Overall, this study demonstrates the potential of soliton macromolecules or polyskyrmionomers as a new form of metamaterials. The unique properties and behaviors of these structures could open up new possibilities in designing solitonic meta matter and exploring applications in various fields. For this reason I think that the paper must be accepted for publication in its current form.

Authors:

We appreciate the Referee #1's recommendation for publication.

Report of Referee 2:

I believe that the paper is a significant step forward in the physics of nonlinear waves. The reported effects are of interest not only for the researches working in the field of liquid crystals or magnetic systems but for wider physical community. Thus I think that the paper can be published in Nature Communications.

Authors:

We appreciate the Referee #2's recommendation for publication.

Referee 2:

...However I have a number of critical remarks listed below.

1) The authors make the statement that "However, the possibility of forming soliton macromolecules or metamaterial analogues of polymers with inter soliton binding resembling strong covalent-like chemical bonds has not been considered, while they can be a basis for new forms of meta matter." I suggest to explain in more details why this case is of interest and if there is a fundamental difference between this kind of inter-soliton coupling and the known ones.

Authors:

We thank Referee #2 for this comment. In the revised paper, we slightly expand the discussion to explain further why using topological defects to bind topological solitons is of interest, and also the differences as compared to known types of binding solitons (see the added sentence in the manuscript's introduction).

Referee 2:

2) There is a statement that equation (1) describes over-damped oscillations. If the oscillations are over-damped then, probably, the equation can be reduced to the first order equation describing the relaxation of the system to the equilibrium state.

Authors:

We thank Referee #2 for this comment. Yes, we also can use the first order equation to describe the motion; however, the respective fitting results are not as good as those provided by Eq. (1), which shows a clear $1/N$ dependence for the effective damping coefficient and eigenfrequency.

Referee 2:

3) This question is related to the previous one. Can equation (1) be understood as the equation for the amplitude of a mode (in the context for the lowest mode amplitude) of the coupled beads described by something like

$$\ddot{x}_n + \gamma \dot{x}_n + \sigma(2x_n - x_{n+1} - x_{n-1}) = 0 ?$$

Then, indeed, the eigenfrequency of the lowest mode scales as $1/N$ where N is the number of the beads. However I do not quite understand the reason why this scaling works for γ .

Authors:

A dynamical equation similar to one proposed by the Reviewer will be relevant for the study for intrinsic wave excitations in the chains of interacting solitons. This is an interesting possibility which should be a subject of a separate work, as it would require detailed comparison with additional experimental data, and an accurate identification of an effective potential of the interaction between adjacent solitons.

Referee 2:

4) I believe that it has to be explained in the text what is the difference between gas-like and liquid-like states (lines 192-200)

Authors:

Thank you for this comment. We referred to “liquid-like states” to describe the system with high density of the polyskyrmionomers. However, because all states are not dense enough to show evident liquid-like properties, in the revised paper we have removed “liquid-like states”, to avoid misunderstanding.

Referee 2:

5) I have to complain on the quality of the figures. The important elements are often so small that they can barely be seen. For example in Fig.1 it is difficult to understand

where some arrows are directed to. Also, I would say that the figures contain too many panels; they are very densely packed, messy and difficult to read. I believe that the figures captions can also be improved.

Authors:

Following this suggestion, we have improved the quality of figures (in particular, by making different component larger) and respective captions in the revised manuscript.

Report of Referee 3:

The presented research is novel and confirmed by the series of high-quality experimental verifications, like 3D nonlinear imaging, undertaken to present the director's spatial distribution in analyzed samples. Also noteworthy is the achievement of high agreement with numerical simulations. This interesting work fosters a better understanding of light-matter interactions in complex pre-engineered optical structures. The manuscript is generally well-written. I think that this work has the potential to be published in Nature Communications Journal.

Authors:

We appreciate the Referee #3's recommendation of our manuscript for publication.